# One Year of Lung Ultrasound in Children with SARS-CoV-2 Admitted to a Tertiary Referral Children’s Hospital: A Retrospective Study during 2020–2021

**DOI:** 10.3390/children9050761

**Published:** 2022-05-23

**Authors:** Anna Maria Musolino, Valentina Ferro, Maria Chiara Supino, Elena Boccuzzi, Simona Scateni, Serena Sinibaldi, Laura Cursi, Paolo Maria Salvatore Schingo, Antonino Reale, Andrea Campana, Massimiliano Raponi, Alberto Villani, Paolo Tomà

**Affiliations:** 1Pediatric Emergency, Department of Emergency and General Pediatrics, Bambino Gesù Children’s Hospital, IRCCS, 00165 Rome, Italy; amcaterina.musolino@opbg.net (A.M.M.); mariachiara.supino@opbg.net (M.C.S.); elena.boccuzzi@opbg.net (E.B.); simona.scateni@opbg.net (S.S.); antonino.reale@opbg.net (A.R.); 2Pediatric Unit, Department of Emergency and General Pediatrics, Bambino Gesù Children’s Hospital, IRCCS, 00050 Palidoro, Italy; serena.sinibaldi@opbg.net (S.S.); andrea.campana@opbg.net (A.C.); 3Immunology and Infectious Disease Unit, Bambino Gesù Children’s Hospital, IRCCS, 00165 Rome, Italy; laura.cursi@opbg.net; 4Department of Imaging, Bambino Gesù Children’s Hospital, IRCCS, 00165 Rome, Italy; pmsalvatore.schingo@opbg.net (P.M.S.S.); paolo.toma@opbg.net (P.T.); 5Medical Direction, Bambino Gesù Children’s Hospital, IRCCS, 00165 Rome, Italy; massimiliano.raponi@opbg.net; 6General Pediatrics Unit, Department of Emergency and General Pediatrics, Bambino Gesù Children’s Hospital, IRCCS, 00165 Rome, Italy; alberto.villani@opbg.net

**Keywords:** COVID-19, lung, ultrasound, children, severity, B-lines

## Abstract

During the COVID-19 pandemic, the lung ultrasound (LU) turned out to be a pivotal tool to study the lung involvement in the adult population, but the same was not well evaluated in children. We detected the LU patterns through an integrated approach with clinical–laboratory features in children hospitalized for COVID-19 in relation to the temporal trend of the Italian epidemic. We conducted a retrospective study which took place at a pediatric tertiary hospital from 15 March 2020 to 15 March 2021. We compared the characteristics of the initial phase of the first COVID-19 year—in the spring and summer (15 March–30 September 2020)—and those of the second phase—in the autumn and winter (1 October 2020–15 March 2021). Twenty-eight patients were studied both in the first and in the second phase of the first COVID-19 year. The disease severity score (DSS) was significantly greater in the second phase (*p* = 0.015). In the second phase of the first COVID-19 year, we detected a more significant occurrence of the following LU features than in the first phase: the irregular pleural line (85.71% vs. 60.71%; *p* = 0.035), the B-lines (89.29% vs. 60%; *p* = 0.003) and the several but non-coalescent B-lines (89.29% vs. 60%; *p* = 0.003). The LU score correlated significantly with the DSS, with a moderate relationship (r = 0.51, *p* < 0.001). The combined clinical, laboratory and ultrasound approaches might be essential in the evaluation of pulmonary involvement in children affected by COVID-19 during different periods of the pandemic.

## 1. Introduction

Since the beginning of the COVID-19 pandemic, numerous efforts have been made to establish guidelines for the diagnosis, treatment and prevention in children [1,2]. Of great concern is which imaging technique is most appropriate to assess lung involvement in pediatric patients. Computed tomography (CT) has been a powerful imaging investigation for supporting the diagnosis and management of adults with COVID-19 infection, especially because of its higher diagnostic sensitivity in the initial disease process [3,4]. Because of how many dark points of this unknown disease there are, especially at the initial phase of the outbreak, CT was often used in the clinical management and diagnosis of children in China [5]. However, recent recommendations from the American College of Radiology assert that CT should not be used as a screening or first-line diagnostic tool for patients with suspected COVID-19 [6]. In fact, due to high ionizing radiation exposure, CT should be avoided or used with caution in the pediatric population [7], limiting the use of CT scans only to children with severe disease [8,9].

Chest X-ray (CXR), which was employed in the first cases of COVID-19, is less specific for the disease, is characterized by a lower diagnostic specificity in the detection of the early pathologic alterations of the infection as the “ground glass” opacities and is not recommended as the gold standard for imaging investigations [10,11].

In recent years, the point of care ultrasound has drawn increasing attention as a useful diagnostic tool in children because of its undeniable series of advantages: it does not expose patients to radiation and therefore allows for a safe and repeatable assessment of the lung involvement in the disease process and a monitoring of the response to the treatment; contrast administration and sedation are not required; it can be performed at the bedside, especially in critically ill children; and it allows for real time visualizations in many scans [12]. Some studies have discussed the role of the LU in the detection of lung alterations in children [10,13] and of neonates with COVID-19 [8] (even in asymptomatic cases), and similar LU findings have been reported in adults [13,14]. Nevertheless, these studies were limited and involved a low number of patients.

To the best of our knowledge, there are still no studies assessing the variation in LU characteristics in combination with clinical and laboratory findings during different times of the pandemic. The first purpose of this study was to analyze the LU patterns through a combined clinical–laboratory approach in the pediatric population hospitalized for COVID-19 infection in relation to the temporal trend of the epidemic. The second purpose was to analyze the relationship between the extent of the lung pathologic process detected by LU and the severity of the disease in relation to the temporal trend.

## 2. Materials and Methods

### 2.1. Study Sample

This was a retrospective study conducted at a pediatric tertiary referral hospital from 15 March 2020 to 15 March 2021. We included patients aged under 18 years old admitted to the hospital with a positive real-time polymerase chain reaction (PCR) on a nasopharyngeal swab and who had undergone LU examination within 12 h of admission.

We compared the characteristics of two periods of the pandemic outbreak: the first one in the spring and summer (15 March–30 September 2020) and the second one in the autumn and winter (1 October 2020–15 March 2021). For each patient, the following data were gathered: demographic information (age and sex), clinical presentation and medical history (underlying disease, fever, respiratory symptoms and extra pulmonary symptoms), vital signs at the time of hospitalization (such as oxygen saturation (SpO2)) and respiratory rate (RR) percentile according to a population-based reference value [15]. Other variables were the requirement of oxygen support and/or the need for a transfer to the pediatric intensive care unit (PICU). For each patient, we used a disease severity score (DSS) described by Parri et al. [16] and based on the adapted classification by Dong et al. [17], as reported in Table 1.

To determine the DSS, each patient underwent CXR. We also described the laboratory tests including C-Reactive Protein (CRP) (mg/dl), Ferritin (ng/mL), White Cell Count (×10^3^/L), Neutrophils (×10^3^/L), Lymphocytes (×10^3^/L) Platelets (10^3^/L), Hemoglobin (g/dL), PT-INR (PT-prothrombin time- and INR-International Normalized Ratio), PTT second (Partial Thromboplastin Time) and Fibrinogen (mg/dL).

### 2.2. LU performance

The performance of LU is an integral routine of the clinical and physical assessment of children with respiratory manifestations at our hospital, as approved by the ethics committees of our institution. In order to reduce the infectious exposure risk of the health-care providers, the LU was performed with a wireless pocket device connected to a probe which was placed in single-use plastic covers (Sonosite iViz), according to a previous, well-described technique [18,19].

The pediatricians with standard adequate training in LU and with at least 3 years of experience in the procedure performed the examination. The LU was conducted within 12 h of admission using a 6–10 MHz lineal probe and through screening the lung fields according to the procedure proposed by Soldati et al. [20].

We evaluated the presence of pleural irregularities, subpleural consolidations, B-lines and pleural effusions. The B-lines were categorized into several but non-coalescent B-lines (Figure 1) and several-coalescent B-lines, the latter also known as “white lung” (Figure 2 and Figure 3).

The lung pattern was classified according to the score proposed by Soldati et al. [20]. The classification includes: score 0—normal sliding, a regular pleural line and A-lines with fewer than 2 B-lines; score 1—a pleural line indented with multiple well-defined B-lines; score 2—a broken pleural line associated with dark areas and consolidation areas; score 3—large and multiple patches of white lung. LU was performed for patients in the sitting position, and 14 areas (3 posterior, 2 lateral, and 2 anterior) were scanned. Therefore, we summarized the lung ultrasound score (LUS) in each area.

### 2.3. Ethics

This study was approved by the Ethics Committee of Bambino Gesù Children’s according to the Declaration of Helsinki (as revised in Seoul, Korea, October 2008) (number protocol 2517_OPBG_2021, approved June 9, 2021).

### 2.4. Statistical Analysis

A Statistical analysis was performed using the software STATA/IC 14.2 version 2017, by StataCorp, College Station, TX, USA. We used the Kolmogorov–Smirnov test to assess the normality of the data distribution. The continuous variables were reported as the mean ± standard deviation (SD) or the median and interquartile range (IQR), as appropriate. The categorical variables were reported as frequencies and percentages (%).

The categorical variables were analyzed by Chi Square. Fisher’s exact tests were implemented when the expected frequency counts were low (<5). For continuous variables, we used Student’s test in situations of normality; otherwise, non-parametric equivalents (Mann–Whitney U) were applied. The Pearson correlation or Spearman rank correlation tests were calculated to analyze the correlation between the LUS score and the DSS and the laboratory data. Generally, a correlation of |r| = 0.0 to 0.2 indicates a very weak association or no association; |r| = +0.2 to 0.4 indicates a weak association; |r| = +0.4 to 0.6 indicates a moderate association; |r| = +0.6 to 0.8 indicates a strong association; |r| = +0.8 to 1.0 indicates a very strong association; |r| = +1 indicates a perfect positive association

We did not perform the logistic regression to explore the relationship between the LUS characteristics and the period of the COVID-19 epidemic outbreak in children hospitalized for COVID-19 infection, because, due to the small sample size, a regular logistic regression is not advisable.

## 3. Results

A total of 56 patients with a median age of 135.5 months (IQR: 67–179.5 months) who were affected by COVID-19 infection were included in the study. In total, 28 patients were seen in the first COVID-19 period group, and 28 patients were seen in the second COVID-19 period group. In Table 2, the demographic, clinical, laboratory and LU findings of the two groups are summarized.

From a clinical viewpoint, in the second COVID-19 period, the patients presented a higher rate of underlying diseases (42.86% versus 17.86%; *p* = 0.042) and a higher DSS (2 [0,1,2] versus 0 [0,1]; *p* = 0.015) than the patients in the first COVID-19 period.

Considering the laboratory data, ferritin and INR were significantly higher in the second period (*p* = 0.007 and *p* = 0.004, respectively), whereas the levels of lymphocytes and platelets were significantly higher in the first period (*p* = 0.01 and *p* = 0.012, respectively).

Among LU pathological features, we observed that the occurrence of the irregular pleural line was seen more frequently in the second period (85.71% vs. 60.71%; *p* = 0.035). The presence of B-lines was significantly more common in children in the second period (89.29% vs. 60%; *p* = 0.003), and the presence of several but non-coalescent B-lines was also significantly more frequent in the second period (46.43% versus 10.71%; *p* = 0.003). The sub-pleural consolidation was another ultrasound finding that was described more frequently in the second COVID-19 period (28.57% versus 10.71%; *p* = 0.04).

No other significant difference was highlighted in the presence of white lung and pleural effusion. However, the only three cases of pneumothorax were registered during the second period.

Finally, the LUS scores were significantly higher in the second period (2 [1,2,3,4] versus 0 [0,1,2,3]; *p* = 0.011).

We analyzed the correlation between the LUS score and the clinical and laboratory findings (Table 3).

We found that the children who had higher SpO2 at admission presented lower LUS scores, with a moderate relationship (r = −0.43; *p* = 0.01). Importantly, the LUS correlated significantly with the DSS, with a moderate relationship (r = 0.51, *p* < 0.001). We noted that, as the PTT value went up, the LUS score went down, but the relationship was weak (r = −0.32; *p* = 0.05).

We also analyzed the correlation between DSS and the clinical and laboratory findings (Table 4).

We observed a positive correlation between the age of children and the DSS: as the children got older, the DSS increased, but with a weak association (r = 0.35; *p* = 0.03). In addition, we reported a significant correlation between the DSS and two clinical parameters. In detail, we noted that the children who had higher SpO2 at admission presented lower DSS values, with a moderate association (r = −0.52; *p* = 0.0008), and the children who had higher RR presented higher DSS values, with a moderate association (r = 0.44; *p* = 0.006).

## 4. Discussion

Although the COVID pandemic provided the opportunity to appreciate how LU is a powerful tool in the hands of clinicians, allowing for the triage and the classification of disease severity in adults, the same was not well described in children, probably due to the extremely low number of symptomatic patients.

To the best of our knowledge, this is the first retrospective study that analyzes LU patterns in combination with the clinical–laboratory profiles of children hospitalized for COVID-19 infection in relation to the temporal trend of the Italian epidemic. In the present study, the severity of the children was higher in the second period (autumn–winter) than in the first period (spring–summer), as well as the LUS. In addition, the LUS correlated significantly with the DSS.

Among the ultrasound findings, the presence of the irregular pleural line and of B-lines occurred more significantly in the second period, as well as the presence of several but non-coalescent B-lines and of sub-pleural consolidations, but no significant difference was highlighted in the cases of white lung and pleural effusion. However, only three cases of pneumothorax were registered during the second period. A previous study conducted in our institution during the first period (from 27 March to 1 June, 2020) showed that the B-lines were the most common finding, while sub-pleural consolidations, white lung and pleural effusions were relatively rare [21]. In addition, the B-lines turned out to be significantly associated with moderate (more than mild) severity disease [21]. The major frequency of the B-lines and of the irregular pleural line and the more elevated presence of the several B-lines might be considered as markers of the severity of the disease; in fact, the occurrence of these findings was significantly more common during the second period, when patients reported a higher DSS. Even in the adult population, LU patterns have been seen to be associated with the severity of diseases and the disease changes [22,23]. In fact, a major frequency of LU findings has been described in cases of severe disease. The irregular pleural line and the subpleural consolidation increased from 40% and 27.7%, respectively, in cases of mild disease to 85.7% and 66.2% in cases of severe disease in adults. Likewise, wide or coalescent B-lines increased from 15.6% to 45.1% [23].

An important result of our study was the significant correlation between disease severity and the LUS. The relation between the LUS and the severity of disease has been well documented in the adult population [24,25,26]. On this topic, there are still few studies on the pediatric population, and they consist of a small sample of pediatric patients. Guitar et al. [16] recruited patients who were divided into two groups depending on the presence of respiratory symptoms, and the LU results were categorized into four degrees [27] according to the score of Soldati et al. [20]. Most of them showed a score of 2, and ten patients presenting non-respiratory symptoms had a score from 0 to 2, whereas three patients who were diagnosed with multisystem inflammatory syndrome had a score of 2. In this study, the Soldati score seemed to be applied in general and not for each field lung; therefore, their data might be reductive in the assessment of the LUS.

The second period was characterized by a significantly higher DSS as well. Studies conducted in the adult population highlight that the number of cases with a severe disease was smaller in the second wave [28,29,30]. However, a lot of studies have reported that children and young people were significantly less affected than adults during the first wave, considering the frequency of cases, the disease severity, the hospital admissions and the mortality [31,32,33,34]. Fewer studies have compared the different waves of the pandemic in children. In the United Kingdom, a study did not find increased disease severity in the second wave compared with the first wave [35]. This difference might reflect the higher frequency of patients with underlying diseases admitted to a hospital during the second period in our study. During the first period, these patients might have been protected by factors that changed with the emergence of the second period: most Italian schools were closed during the first pandemic wave in the spring of 2020, as well as in the summer; some children were identified as extremely clinically vulnerable and were advised to shield themselves from all non-essential contact; there was an emergence of new variants such as the alpha variant (B.1.1.7) during the second period [35]. In addition, we considered the epidemiology of the other co-infections that may have changed the disease severity.

We reported a more significant decrease in lymphocytes in the second period than in the first period. In the adult population, the lymphopenia is clearly associated with disease severity [36]. Nevertheless, in children with severe COVID-19, nearly equal frequencies of increased and decreased lymphocyte counts were reported, with most children showing normal counts [37]. The immaturity of the immune system in young children might explain the differences in viral susceptibility or the response to infection and might illustrate the differences in the laboratory profiles noted in the pediatric population compared with the adult population with COVID-19 infection [38]. Similarly, we noted a significant increase in INR and a significant decrease in the platelet count in the second period. A trend of elevated PT was found in children with severe COVID-19, although this variable has not been consistently evaluated in the studies [38]. Likewise, thrombocytopenia has predominantly been seen in severe and critical cases [39]. Finally, in the second period, a more elevated ferritin was reported than in the first period, and even then, it might correlate with higher DSS, because the severe clinical course has been seen to be associated with high levels of ferritin [40].

Our study has a lot of limitations. Firstly, it was a single-center study conducted at a large academic hospital in Rome; therefore, the results may not be generalizable across Italy. Secondly, the applicability of LU might be limited by it being an operator-dependent technique; however, we did not test an inter intra-observer agreement. This can be minimized considering that experienced pediatricians performed the LU using a standardized procedure. In addition, our sample size was small, and the number of cases per predictor in some cases was <10. Due to this last limitation, we were not able to identify, among the LU and clinical–laboratory findings, the predictive and prognostic factors of severe disease by performing a multivariable logistic regression; rather, we were only able to indicate whether the LUS was highly related to the severity of the disease.

## 5. Conclusions

Our study shows two important focal points: The first is the potential role of LU in the evaluation of pulmonary involvement in children with COVID-19 during different temporal phases of the pandemic in an integrated, clinical–laboratory work-up.

Secondly, there is a significant correlation between the extent of the lung pathologic process detected by LUS and the severity of the disease in relation. From this perspective, in the future, LUS might be a powerful tool in the hands of pediatricians for the risk stratification of children infected with COVID-19 and for clinical decision processes.

## Figures and Tables

**Figure 1 children-09-00761-f001:**
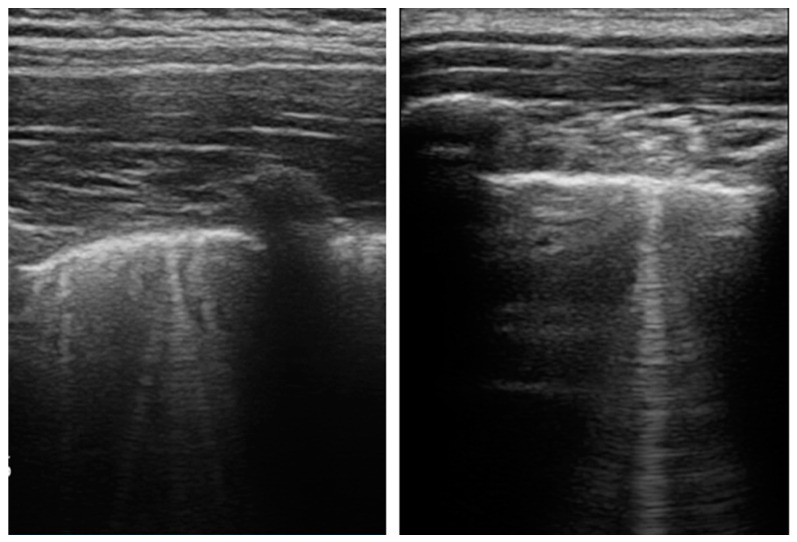
The presence of several but non-coalescent B-lines in the pulmonary parenchyma of a child with COVID-19 infection.

**Figure 2 children-09-00761-f002:**
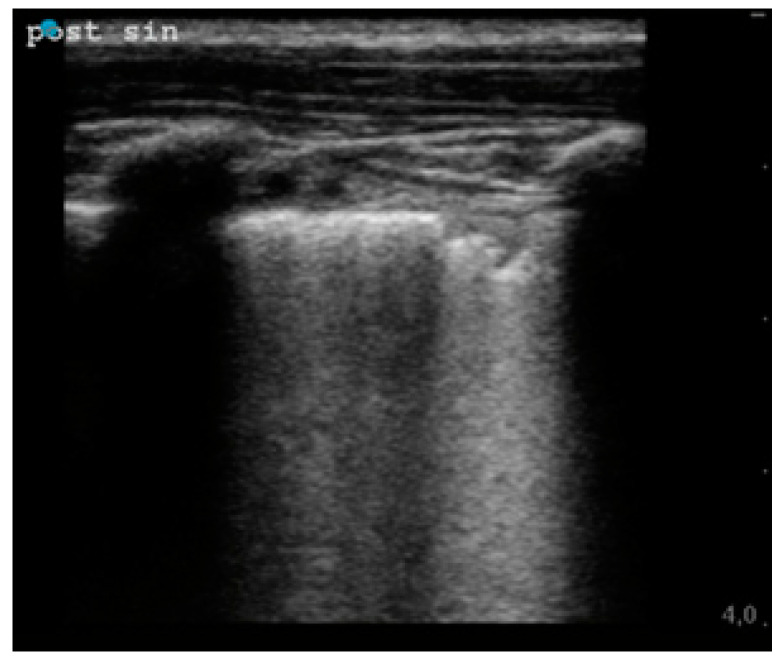
The presence of coalescent B-lines (white lung) associated with the irregular pleural line in the pulmonary parenchyma of a child with COVID-19 infection. The blue point on the left side of the screen, as it is viewed, corresponds to the side of the probe marked with an indicator.

**Figure 3 children-09-00761-f003:**
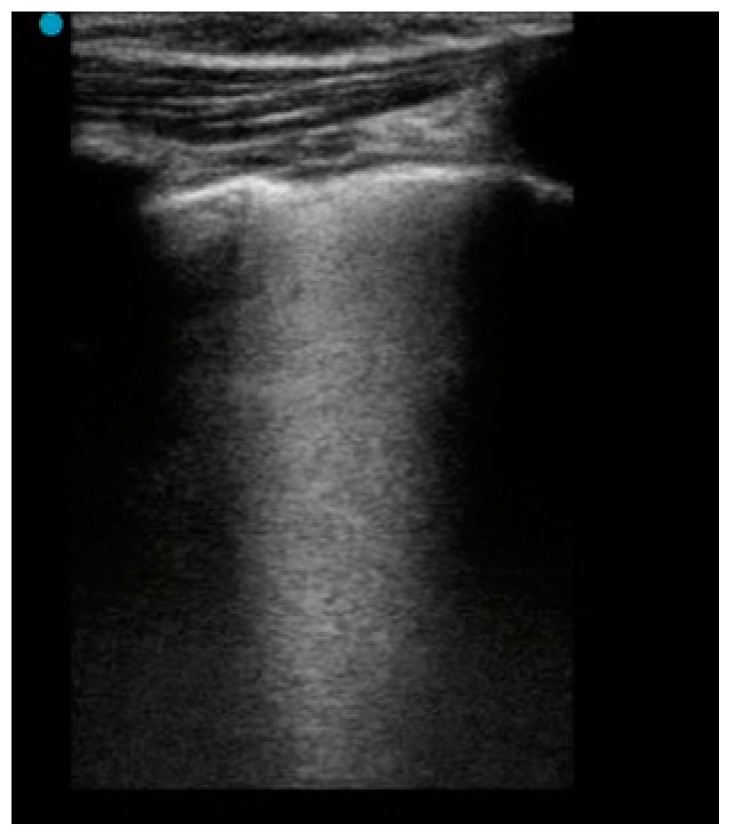
The presence of coalescent B-lines (white lung) in the pulmonary parenchyma of a child with COVID-19 infection.

**Table 1 children-09-00761-t001:** Disease severity score described by Parri (adapting a previous classification) [16,17].

**Asymptomatic: All the following must be present** No signs or symptoms AND negative chest X-ray (CXR) AND the absence of criteria for other cases
**Mild: Any of the following (AND absence of criteria for more severe cases)** Symptoms of upper respiratory tract infection AND the absence of pneumonia at CXR
**Moderate: All the following (AND absence of criteria for more severe cases)** Cough AND sick appearing or pneumonia at CXR
**Severe: Any of the following (AND absence of criteria as for critical case)** 3. Oxygen saturation less than 92% 4. OR difficult breathing or other signs of severe respiratory distress (apnea, gasping, head nodding) 5. OR need for any respiratory support
**Critical: Any of the following** Patient in the intensive care unit OR intubated OR multiorgan failure OR shock, encephalopathy, myocardial injury or heart failure, coagulation dysfunction, acute kidney injury

**Table 2 children-09-00761-t002:** Clinical, laboratory and LUS findings in the epidemic periods of the outbreak in children hospitalized for COVID-19 infection.

Characteristics	First COVID-19 Period n = 28	Second COVID-19 Period n = 28	*p* Value
Sex, n (%) Female Male	13 (46.43) 15 (53.57)	9 (32.14) 19 (67.86)	0.2
Age (months), median (IQR)	113.5 (73.5–165.5)	168.5 (47.5–204.5)	0.08
Underlying disease, n (%)	5 (17.86)	12(42.86)	0.042
Fever, n (%)	17 (60.71)	19 (67.86)	0.57
Respiratory symptoms, n (%)	11 (39.29)	16 (57.14)	0.28
Other symptoms, n (%)	13 (46.43)	16 (57.14)	0.46
Respiratory distress, n (%)	4 (14.29)	4 (14.29)	1
SpO2 at admission, median (IQR)	99 (98–99)	98 (96–98)	0.26
RR age percentile at admission, median (IQR)	50 (10–50)	50 (10–50)	1
Oxygen need, n (%)	0	3 (10.71)	0.118
Intensive recovery need, n (%)	2 (7.14)	4 (14.29)	0.33
Disease severity score, median (IQR)	0 (0–1)	2 (0–2)	0.015
CRP (mg/dL), median (IQR)	0.075 (0.04–0.22)	0.185 (0.035–0.72)	0.36
White cell count (×10^3^/L), median (IQR)	5505 (4465–8460)	6254.5 (4370–9080)	0.8
Ferritin (ng/mL), median (IQR)	74 (40–112)	173.5 (83.5–570)	0.007
Neutrophils (×10^3^/L), median (IQR)	2260 (1890–3825)	2810 (1980–5710)	0.18
Lymphocytes (×10^3^/L), median (IQR)	2280.5 (1870–3930)	1470 (835–2490)	0.01
Platelets (10^3^/L), median (IQR)	253,500 (220,500–306,500)	217,000 (145,500–260,500)	0.012
Hemoglobin (g/dL), median (IQR)	13 (12.2–13.8)	13.25 (11.7–14.85)	0.30
INR, median (IQR)	1 (0.95–1.06)	1.105 (1.02–1.17)	0.004
PTT sec, median (IQR)	30.95 (27.7–32.6)	31.2 (28.4–34)	0.96
Fibrinogen (mg/dL), median (IQR)	309.5 (261–376)	309 (237–358)	0.95
Irregular pleural line, n (%)	17 (60.71)	24 (85.71)	0.035
Irregularity pleural line in multiple segments, n (%)	13 (46.43)	9 (32.14)	0.27
Bilateral location of irregularity pleural line, n (%)	8 (28.57)	11 (39.29)	0.39
B-Lines, n (%)	15 (60.00)	25 (89.29)	0.003
B-Lines in multiple segments, n (%)	6 (21.43)	6 (21.43)	1
Bilateral location of B-Lines, n (%)	4 (14.29)	10 (35.71)	0.06
Several, non-coalescent B-Lines, n (%)	3 (10.71)	13 (46.43)	0.003
Several, coalescent B-Lines (white lung), n (%)	2 (7.14)	7 (25.00)	0.07
Sub-pleural consolidation, n (%)	2 (7.14)	8 (28.57)	0.04
Pleural effusion, n (%)	3 (10.71)	2 (7.14)	0.63
Pneumothorax, n (%)	0	3 (10.7)	0.12
LUS score, median (IQR)	0 (0–3)	2 (1–4)	0.011

**Table 3 children-09-00761-t003:** The correlation between the LUS score and the disease severity score with the clinical and laboratory parameters in children hospitalized for COVID-19 infection.

Characteristic	Lung Ultrasound Severity (LUS) Score	
	Correlation Coefficient (r)	*p* Value
Age (months)	0.22	0.18
SpO2, at admission	−0.43	0.01
RR age percentile, at admission	0.23	0.17
Disease Severity Score (DSS)	0.51	<0.001
CRP (mg/dL)	0.04	0.82
White cell count (×10^3^/L)	−0.24	0.15
Neutrophils (×10^3^/L)	−0.07	0.63
Lymphocytes (×10^3^/L)	−0.22	0.19
Ferritin (ng/mL)	0.09	0.58
Platelets (10^3^/L)	−0.12	0.49
Hemoglobin (g/dL)	−0.03	0.86
INR	−0.08	0.65
PTT sec	−0.32	0.05
Fibrinogen (mg/dL)	0.06	0.75

**Table 4 children-09-00761-t004:** The correlation between DSS and the clinical and laboratory parameters in children hospitalized for COVID-19 infection.

Characteristic	Disease Severity Score (DSS)	
	Correlation Coefficient (r)	*p* Value
Age (months)	0.35	0.03
SpO2, at admission	−0.52	0.0008
RR age percentile, at admission	0.44	0.006
Lung Ultrasoud Severity (LUS) Score	0.51	<0.001
CRP (mg/dL)	0.18	0.29
White cell count (×10^3^/L)	0.08	0.64
Neutrophils (×10^3^/L)	0.25	0.14
Lymphocytes (×10^3^/L)	−0.22	0.18
Ferritin (ng/mL)	0.19	0.26
Platelets (10^3^/L)	−0.2	0.24
Hemoglobin (g/dL)	0.09	0.59
INR	−0.16	0.33
PTT sec	−0.28	0.09
Fibrinogen (mg/dL)	0.13	0.45

## Data Availability

The data presented in this study are available on request from the corresponding author.

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
