# Peer review of "One Year of Lung Ultrasound in Children with SARS-CoV-2 Admitted to a Tertiary Referral Children’s Hospital: A Retrospective Study during 2020–2021"

_children, 2022, doi:10.3390/children9050761_

Round 1

Reviewer 1 Report

The analysis of the two groups has more an epidemiological value of the health condition in the pediatric population with Covid infection. Therefore, a recommendation should be considered.

By using CXR as the imaging evaluation method in analyzing the disease severity score (DSS), a practical utility for the clinicians would be to compare the radiographic method with the ultrasound method. This would subscribe to the information that the authors mentioned in the body of the article, according to which lung ultrasound has the benefits of assessing lung damage in children.

A comparative analysis between the CXR and lung ultrasound in total 56 patients (as a single group), correlated with clinical and biological data could increase the value of the statistical analysis of the group.

Author Response

We agree with your consideration, in fact in the table 3 we reported the correlation between the disease severity score (DSS) on the presence of chest X-ray lesions with those of the lung ultrasound score (LUS) and we found, importantly, that the LUS correlated significantly with the DSS with a moderate relationship (r=0.51, p<0.001), that to say, as the value of the lung ultrasound score increase, the value of the disease severity score increases. In the table 3 we also reported the correlation of LUS with the other continuous variables, both clinically and biologically.

Newly, we added the table 4,  in which we reported the results of the correlation of the DSS with the same continuous variables, both clinically and biologically.

Reviewer 2 Report

Good material to try to investigate the US utility in children COVID-19 where the prevalence and the severity was low.

In Table 1, row 4 the words "entry 2 data data1" should be erased.

The last sentence of the conclusions talks about COVID-10, it must be corrected.

Author Response

We corrected mistakes in the table 1 and in the conclusion.

Round 2

Reviewer 1 Report

The revised form in the result and discussion brings a more structured and easy to read and understand the work of the authors with a consistency of the data and the correlations.